# Dimethyloxalylglycine (DMOG), a Hypoxia Mimetic Agent, Does Not Replicate a Rat Pheochromocytoma (PC12) Cell Biological Response to Reduced Oxygen Culture

**DOI:** 10.3390/biom12040541

**Published:** 2022-04-03

**Authors:** RuoLi Chen, Mohammad Alkataan Ahmed, Nicholas Robert Forsyth

**Affiliations:** 1School of Pharmacy and Bioengineering, University of Keele, Newcastle Under Lyme, Staffordshire ST5 5BG, UK; moh1977729@gmail.com (M.A.A.); n.r.forsyth@keele.ac.uk (N.R.F.); 2College of Pharmacy, University of Mosul, Mosul 41002, Iraq

**Keywords:** hypoxia, PC12 cells, HIF, proliferation, differentiation, cell cycle, mitochondria

## Abstract

Cells respond to reduced oxygen availability predominately by activation of the hypoxia-inducible factor (HIF) pathway. HIF activation upregulates hundreds of genes that help cells survive in the reduced oxygen environment. The aim of this study is to determine whether chemical-induced HIF accumulation mimics all aspects of the hypoxic response of cells. We compared the effects of dimethyloxalylglycine (DMOG) (a HIF stabiliser) on PC12 cells cultured in air oxygen (20.9% O_2,_ AO) with those cultured in either intermittent 20.9% O_2_ to 2% O_2_ (IH) or constant 2% O_2_ (CN). Cell viability, cell cycle, HIF accumulation, reactive oxygen species (ROS) formation, mitochondrial function and differentiation were used to characterise the PC12 cells and evaluate the impact of DMOG. IH and CN culture reduced the increase in cell numbers after 72 and 96 h and MTT activity after 48 h compared to AO culture. Further, DMOG supplementation in AO induced a dose-dependent reduction in the increase in PC12 cell numbers and MTT activity. IH-cultured PC12 cells displayed increased and sustained HIF-1 expression over 96 h. This was accompanied by increased ROS and mitochondrial burden. PC12 cells in CN displayed little changes in HIF-1 expression or ROS levels. DMOG (0.1 mM) supplementation resulted in an IH-like HIF-1 profile. The mitochondrial burden and action potential of DMOG-supplemented PC12 cells did not mirror those seen in other conditions. DMOG significantly increased S phase cell populations after 72 and 96 h. No significant effect on PC12 cell differentiation was noted with IH and CN culture without induction by nerve growth factor (NGF), while DMOG significantly increased PC12 cell differentiation with and without NGF. In conclusion, DMOG and reduced oxygen levels stabilise HIF and affect mitochondrial activity and cell behaviour. However, DMOG does not provide an accurate replication of the reduced oxygen environments.

## 1. Introduction

Neural stem cells (NSCs) and neural progenitor cells (NPCs) are capable of both self-renewal and differentiation into the principal cell types of the nervous system [1]. NSC and NPC self-renewal and differentiation are tightly controlled by the cellular microenvironment or “niche”. For instance, high glucose concentrations inhibit NSC differentiation through oxidative endoplasmic reticulum stress [2], while physiological oxygen maintains stemness [3]. Oxygen levels not only maintain the undifferentiated state, but also modulate the proliferation, quiescence and fate commitment of NSCs [1]. Under physiological conditions, oxygen levels vary from one tissue to another, ranging from 0.2% to 10%, which is lower than the atmospheric oxygen level of 20.9% [4]. Normoxia is defined as the oxygen partial pressure (pO_2_) that an organism, tissue or niche within tissue routinely experiences. Any reduction in this routine normoxia, disrupting bioavailability levels of O_2_, is called hypoxia. Hypoxia is defined as a state of reduced O_2_ supplied to cells or tissue that results in a reduction in ATP production in one state or condition compared with another [5]. As such, the highly relative term “hypoxic status” differs from one tissue, niche or cell type to another [4].

Cells respond to hypoxia through an evolutionary mechanism that is regulated, at least in part, by hypoxia-inducible factor (HIF), which is the master regulator of oxygen homeostasis and is critical for adaptation to hypoxic insult [6]. HIF is an alpha/beta heterodimeric transcription factor that regulates the expression of hundreds of genes in a cell-dependent manner. In humans, there are three isoforms: HIF-1, -2 and -3 [7]. HIF-1 and -2 are structurally similar and share some common transcriptional targets (e.g., *Slc2A1* and *VEGF*) but also regulate distinct subsets of genes and elicit different cellular fates [8,9]. HIF-1 appears to play a significant role in the regulation of many metabolic activities such as glucose haemostasis and lipid and amino acid metabolism [4]. HIF-2 promotes angiogenesis, cell division and tissue regeneration by regulating the expression of *EPO*, *Cyclin D1* and *Oct4* [10]. HIF-1 appears to be involved in adaptation to acute hypoxia whilst *HIF-2* drives the chronic response to hypoxia [11]. By contrast, HIF-3 exists in several alternatively spliced forms and may negatively regulate HIF-1 and -2 [12].

HIF levels in cells are tightly controlled by a group of hydroxylases: three prolyl hydroxylases (PHD1-3) and one asparaginyl hydroxylase. In normoxia, the PHDs signal HIF alpha degradation [13]. In hypoxia, these enzymes do not function as there is a lack of oxygen, one of the substrates for the PHDs. HIFs are therefore stabilised and upregulate hundreds of downstream genes [14]. A number of chemical or biological molecules are able to stabilise HIF and are termed hypoxia mimetic agents (HMAs) [14]. Most HMAs function by targeting PHD, such as iron chelates and analogues of 2-oxoglutarate (2-OG), another substrate for the PHDs [14]. Dimethyloxalylglycine (DMOG), an ester of N-oxalyglycine which has a similar structure to 2-OG, has been shown to stabilise HIF-1 nonspecifically in a number of studies both in vitro and in vivo [15,16,17]. DMOG is commonly used as a reference HMA when developing specific and highly potent HMAs for clinical use [13,15] and was shown to better mimic the transcriptional response to hypoxia than selective PHD inhibitors in MCF-7 cells [13].

The aim of this study was to compare the DMOG-induced responses to those at lower oxygen levels (2% O_2_) through comprehensive interrogation of PC12 cell viability, cell cycle analyses, HIF-1 and -2 accumulation, reactive oxygen species (ROS) formation, mitochondrial function and differentiation. Rat pheochromocytoma (PC12) cells, a neuron-like cell line, have the ability to differentiate, thus providing a means to study NSC growth and differentiation [18,19]. These responses were all characterised against either intermittent hypoxia at 2% O_2_ (IH) or continuous normoxia at 2% O_2_ (CN).

## 2. Methods

### 2.1. Cell Culture

PC12 cells were obtained commercially from Sigma-Aldrich (St. Louis, MO, USA) and were maintained in high-glucose DMEM supplemented with 5% foetal bovine serum, 5% horse serum and penicillin–streptomycin (100 U/mL and 100 µg/mL, respectively) in a humidified 5% CO_2_ and 20.9% O_2_ atmosphere at 37 °C.

PC12 cells were counted either manually with a Neubauer haemocytometer or with a Countess Automated Cell Counting Platform system (Invitrogen, Thermo Fisher Scientific, Waltham, MA, USA). Trypan blue exclusion was used to determine viability.

### 2.2. Alternate Oxygen Level Culture Conditions

Alternate oxygen conditions were established with three distinct cell culture environments. PC12 cells were seeded into a standard air oxygen (AO) cell culture incubator (Panasonic MCO-18AC-EP, Kadoma, Osaka, Japan), a tri-gas incubator (Panasonic MCO-19M-PE, Kadoma, Osaka, Japan) for intermittent hypoxia (IH) (2% O_2_) or a hermetic workstation with controllable oxygen levels (SCI-TIVE, Baker Ruskinn, Bridgend, Glamorgan, UK) for a continuous normoxia (CN) environment (2% O_2_), as reported previously [20].

### 2.3. DMOG Treatment

DMOG (Sigma-Aldrich, St. Louis, MO, USA) was dissolved in ddH_2_O before being utilised as a supplement for PC12 cells in AO with concentrations ranging from 0.1 to 1 mM. Control groups consisted of cells grown in the absence of DMOG.

### 2.4. MTT Assays

Cell metabolic activity was evaluated by a standard colorimetric assay for mitochondrial reductase-catalysed reduction of yellow MTT (3-(4,5-dimethylthiazol-2-yl)-2,5-diphenyltetrazolium bromide) (Sigma-Aldrich, St. Louis, MO, USA) to a purple formazan product. PC12 cells were seeded on poly-D-lysine precoated 96-well plates at a density of 1.2 × 10^4^ cells/well and cultured in “complete” medium. Four hours prior to the completion of the routine culture incubation period, 10 µL of 5 mg/mL MTT solution diluted in Dulbecco’s phosphate-buffered saline (PBS) (Thermo Fisher Scientific, Waltham, MA, USA) was added to the culture medium (final concentration 0.5 mg/mL), and all samples incubated at 37 °C under treatment conditions. At the completion of treatment, the culture supernatant was aspirated, and the formazan crystals formed by surviving cells were solubilised in 50 µL DMSO (Sigma-Aldrich, St Louis, MO, USA) and incubated at 37 °C for a further 10 min. The optical density (OD) value of each well was determined by reading absorbance at 540 nm using a Tecan Infinite M200 PRO microplate reader (Tecan Group Ltd., Reading, UK). The absorbance reading of the background was subtracted from all sample absorbance readings, and the viability of cells for each treatment group was calculated based on Equation (1).
(1)$$\mathrm{MTT}\;\left(\%\;\mathrm{of}\;\mathrm{the}\;\mathrm{control}\right)=\frac{\mathrm{OD}\;\mathrm{value}\left(\mathrm{Experimental}\;\mathrm{group}\right)}{\mathrm{OD}\;\mathrm{value}\left(\mathrm{Control}\;\mathrm{group}\right)}\times 100\%$$

The viability of control cells (complete media in normoxic conditions) was assigned as 100%, while treatment samples were normalised against the control group OD value. Results are expressed as the percentage of cells possessing the ability to reduce MTT.

### 2.5. Hypoxia-Inducible Factor-1α Subunit Expression Analysis

PC12 cells were seeded at 4 × 10^4^ cells /mL and incubated for 24, 48, 72 and 96 h. Following incubation, the cells were fixed with 80% ice-cold methanol (Thermo Fisher Scientific, Waltham, MA, USA) for 5 min and the methanol was removed. After permeabilisation with 0.1% PBS/Tween for 20 min, cells were incubated in a blocking solution consisting of 10% Bovine Serum Albumin (BSA) with 0.3 M glycine (Sigma-Aldrich, St. Louis, MO, USA) with primary monoclonal antibodies for HIF-1α (ab16066) (Abcam, Cambridge, UK) at a concentration of 2 µg/1 × 10^6^ cells and incubated for 30 min at 22 °C. The primary antibody-blocking solution was then removed via centrifugation at 300× *g* for 3 min. The cells were washed with PBS, centrifuged for 3 min at 300× *g* and then incubated with 500 µL of secondary antibody DyLight 488 goat anti-mouse IgG (H + L) (Abcam, Cambridge, UK) (1/500 dilution) for 30 min at 22 °C. An FC500 flow cytometer (Beckman Coulter, Brea, CA, USA) was used to analyse the samples. At least 10,000 events were collected per sample. Data were analysed using Flowing Software (Turku Centre for Biotechnology, Turku, Finland).

### 2.6. Reactive Oxygen Species (ROS) Measurement

ROS were measured by staining cells with ROS-ID Hypoxia/Oxidative Stress Detection Kit (Enzo Life Sciences, Farmingdale, NY, USA) according to the manufacturer’s instruction. Briefly, cells were seeded as described above. After the indicated treatment, the cells were resuspended (5 × 10^5^ cells) in 200 µL of ROS-ID Hypoxia/Oxidative Stress Detection mixture comprising 6 µL of oxidative stress reagent (5 mM) for 30 min at 37 °C. After incubation, cells were washed with PBS (Thermo Fisher Scientific, Waltham, MA, USA) and stored in the dark prior to analysis via Cytomics FC500 flow cytometer (Beckman Coulter, Brea, CA, USA). At least 10,000 events were collected per sample. Data were analysed using Flowing Software (Turku Centre for Biotechnology, Turku, Finland).

### 2.7. Mitochondrial Burden and Action Potential

PC12 cells were seeded at 4 × 10^4^ cells /mL. After the indicated treatment, media were removed and mitochondria were labelled with a mixture of 100 nM MitoTracker Green FM (mitochondrial burden) and 25 nM MitoTracker Red FM (mitochondrial membrane action potential) (Thermo Fisher Scientific, Waltham, MA, USA) in fresh culture medium for 20 min. Cells were then washed with PBS and analysed on a Cytomics FC500 flow cytometer (Beckman Coulter, Brea, CA, USA). A minimum of 10,000 events were collected per sample. Data were analysed with Flowing Software (Turku Centre for Biotechnology). The next step was ensuring that compensation could be performed where dual MitoTracker sample labelling was applied to avoid bleeding across channels. Voltage channels were first set for fluorescence channels using an unstained sample and forward scatter/side scatter was adjusted to clearly delineate the cell population; gating was then applied to exclude dead cells, clumps and debris. Compensation was started from the red fluorochrome stepwise down to the green fluorochrome while checking the compensation in all channels.

### 2.8. Cell Cycle Analysis

PC12 cells were seeded at 4 × 10^4^ cells /mL. Following incubation, cells were centrifuged for 3 min at 300× *g* at 4 °C. The supernatant was aspirated from the pellet of cells and 2 mL of ice-cold 70% ethanol was added slowly to the sample with vortexing. The cells were spun down for 5 min at 500 g, the ethanol was aspirated and the cells were washed with 1 mL PBS twice. Cells were then resuspended in 200 µL propidium iodide (PI) (50 µg/mL) with 50 µL of ribonuclease (100 µg/mL) and incubated in the dark for 30–45 min before having fluorescence measured via a Cytomics FC500 flow cytometer (Beckman Coulter, Brea, CA, USA) with gating applied to exclude dead cells, clumps, debris and doublets. Data were analysed using Flowing Software (Turku Centre for Biotechnology, Turku, Finland).

### 2.9. Differentiation

PC12 cells were plated at a seeding density of 10^4^ cells/mL into 24-well plates coated with collagen type IV. After culturing for 48 h, the media were aspirated and fresh media supplemented with 100 ng/mL nerve growth factor (NGF) (Sigma-Aldrich, St. Louis, MO, USA) were added. Over the next 7–10 days, media were changed every 48 h. Images were recorded using the Nikon Eclipse Ti microscope via a D5-Fil camera on NIS Elements software (Nikon, Shinagawa, Tokyo, Japan).

### 2.10. Data Analysis

The data were expressed as a mean value ± standard error (S.E.M) and were tested for normality using Shapiro–Wilk test. An unpaired two-tailed *t*-test was used for single dataset comparisons against a control. One-way ANOVA with Tukey multiple comparison post hoc test was used to analyse comparisons among multiple groups. GraphPad Prism 7 for Windows version 7.04 (GraphPad Software, Inc., San Diego, CA, USA) was used for the analysis. Values of *p* < 0.05 were considered statistically significant.

## 3. Results

### 3.1. Effect of Oxygen Levels and DMOG on PC12 Cell Count and Metabolic Activity

Under the three oxygen levels, the increase rates of cell numbers were not significantly different over the first 48 h incubation but were reduced after 72 and 96 h incubation in both 2% O_2_ settings (Figure 1A). IH and CN both displayed significantly reduced mitochondrial metabolism at 48 h compared to AO, but no significant differences were noted under the three oxygen levels at other time points tested (Figure 1B). Furthermore, MTT activity values at 72 and 96 h under the three oxygen conditions were significantly lower than the corresponding values at 24 h (Figure 1B).

DMOG displayed dose-dependent effects on the increase in PC12 cell numbers. A dose-dependent reduction in the increase rates of cell numbers was observed at 0.5 mM and 1 mM after 2 days of treatment, as compared to control cells (Figure 2A). When cells were treated with 0.1 mM DMOG, the increase rates of cell numbers increased slightly but nonsignificantly (Figure 2A). These results were confirmed by the MTT cell metabolic activity assay, where a DMOG dose-dependent reduction was observed at all time points, reaching a 49% reduction at 1 mM at 96 h, as compared to control cells (Figure 2B). For further analysis, 0.1 mM DMOG was utilised due to its lack of striking impact on cell proliferation and metabolic activity observed during the 96 h course.

### 3.2. Effect of Oxygen Levels and DMOG on PC12 Cell HIF-1 Expression

IH-cultured cells displayed an immediate upregulation of HIF-1α to 64% after 24 h which was downregulated to 55% by 48 h and maintained thereafter at 49% at 72 h and 55% at 96 h, compared to the consistent baseline expression of 34% in AO (Figure 3). However, CN displayed a consistently low expression profile over the entire time course (30% at 24 h, 43% at 48 h, 33% at 72 h and 33% at 96 h). DMOG (0.1 mM) supplementation in AO significantly upregulated HIF-1 expression (61% at 24 h, 48% at 48 h, 48.8% at 72 h and 43.6% at 96 h) (Figure 3).

### 3.3. Effect of Oxygen Levels and DMOG on ROS Production

There were significant increases in ROS formation after 48 and 72 h of incubation under IH, while no significant changes in CN were observed between 24 and 72 h compared to ROS formation in AO (Figure 4). Cells in AO for 96 h exhibited significantly increased ROS formation. DMOG (0.1 mM) caused a significant reduction in ROS formation (*p* < 0.01) after 24 h but had no significant effect after 48 h, while at 96 h, DMOG reduced the increase in ROS formation levels (Figure 4).

### 3.4. Effect of Oxygen Levels and DMOG on Mitochondrial Burden and Action Potential

There was no significant change in mitochondrial burden during the 96 h incubation in AO. IH significantly elevated mitochondrial burden over 96 h, while there was no change in mitochondrial burden in CN except for a significant reduction at 72 h compared to AO (Figure 5 and Figure 6A). DMOG (0.1 mM) significantly elevated mitochondrial burden at 24 and 96 h compared to AO control, with a significant drop noted at 72 h (Figure 5 and Figure 6A). Under IH and CN, the mitochondrial action potential increased significantly compared to that in AO, while AO-cultured PC12 cells exposed to 0.1 mM DMOG exhibited significantly elevated mitochondrial action potential after 24 h (Figure 5 and Figure 6B).

### 3.5. Effect of Oxygen Levels and DMOG on PC12 Cell Cycle Progression

In comparison to AO, the cell population in G0/G1 phase of the cell cycle was significantly increased after IH and CN culture, but that in the S phase was significantly reduced. The cell population in the G2-M phase was significantly decreased after 24 h in both IH and CN in comparison to AO, with no significant effect being noticed after 48 h (Figure 7). Under the AO culture condition, DMOG did not change the cell population in the G0/G1 phase in comparison to the AO control (Figure 7), while DMOG significantly increased the cell population in the S phase and reduced that in the G2-M phase after 72 and 96 h in comparison to AO control (Figure 7).

### 3.6. Effect of Oxygen Levels and DMOG on PC12 Cell Differentiation Capacity

There was no significant effect caused by the different oxygen culture conditions on the neurite development capacity of PC12 cells without induction by NGF (Figure 8 and Figure 9). After NGF induction, significantly increased neurite outgrowth of PC12 cells was observed in all oxygen culture conditions, with no significant difference among them (Figure 8 and Figure 9). DMOG culture under AO significantly increased the percentage of neurite-bearing cells in comparison to AO with/without NGF (*p* < 0.05) (Figure 8 and Figure 9).

## 4. Discussion

Cells are exposed to different oxygen concentrations in the body regulated by a number of factors, e.g., location, organ vasculature and function [4]. Usually, 2–3% O_2_ is regarded as a physiologically relevant oxygen level, termed “physiological normoxia” or “physioxia” [4]. In this study, we used a continuous control oxygen workstation to provide a continuous 2% “normoxic” oxygen culture (CN), while IH saw cells grown at 2% oxygen in an incubator in which the oxygen level would reset to air oxygen each time the incubator door was opened (for ~1 min). This setting has been applied in human bone-marrow-derived mesenchymal stem cell culture, in which frequent ambient oxygen exposure events caused by incubator door opening resulted in reductions in colony-forming unit fibroblastic yield and widespread transcriptional alterations. [20] We found 2% O_2_ applied either intermittently or constantly significantly reduced cell numbers over the 96 h of continuous culture. Culture at 2% O_2_ significantly reduced MTT activity after 48 h, but MTT activity returned to the baseline value at 72 and 96 h. This is consistent with studies on PC12 cells, where MTT activity (measured by CCK 8) was significantly reduced after 6 and 48 h culture with 1% O_2_ [21,22] and 24 h culture with 3% O_2_ [23]. DMOG expressed dose-related effects on cell proliferation and MTT activities over the 4-day course. This is also consistent with studies in which DMOG showed dose-related reductions in the increase in cell numbers and MTT activity [24,25]. A 0.1 mM concentration of DMOG was chosen for subsequent studies because 0.1 mM DMOG did not significantly change the cell behaviour (i.e., cell number increase rates and MTT activity) (Figure 2), while both 0.5 mM and 1 mM DMOG significantly changed cell behaviour (Figure 2). DMOG acts as a competitive antagonist for the 2-OG cofactors of the PHDs and other 2-OG oxygenases (and other enzymes, such as isocitrate dehydrogenase) [26]. Its effects on cell behaviour could be caused by both HIF-dependent and non-HIF-dependent mechanisms [27,28].

HIFs are rapidly degradable proteins, which can degrade within the first 5 min of exposure to oxygen [29]. The most common methods for detecting HIF alpha proteins include immunohistochemistry, flow cytometry, ELISA or Western blotting. Flow cytometry has been found to be more sensitive than visual immunofluorescence microscopy for the detection of ultralow levels of protein and may facilitate the correct identification of negative cells when moderate or high levels of protein are present in a significant portion of cells [30,31,32,33]. HIF alpha has been detected with flow cytometry by several previous studies [34,35,36]. In this study, we applied flow cytometry to detect HIF-1α and showed significant increases in HIF-1α with the hypoxia treatment and DMOG.

The mitochondria are the main source of endogenous ROS, mainly through electron transport chain complexes I and III [37]. About 95% of endogenous ROS in a normal cell are generated in the mitochondria, while the rest can be formed by a group of enzymes, e.g., nicotinamide adenine dinucleotide phosphate (NADPH) oxidase (Nox), monoamine oxidase, mitochondrial BKCa and mKATP channels, cytochrome b5 reductase and dihydroorotate dehydrogenase [38,39].

Elevated ROS levels in hyperoxic air culture are not surprising given the increased oxygen saturation levels [40]. It has been established that hypoxia increases ROS through the transmission of electrons from ubisemiquinone to molecule oxygen at the Qo site of complex III of the mitochondrial transport chain [4,41]. Ischaemia is also known to induce the generation of ROS by both mitochondria [42] and cytosolic enzymes [43]. Hippocampal neurons have high basic production of ROS, and their ROS production increases rapidly during oxygen and glucose deprivation [39].

HIF-1α elevation expression contributes to ROS formation, specifically the mitochondrial ROS [44] On the other hand, ROS could reduce HIF-1 by upregulation of PHD2 or by donating oxygen in hypoxic conditions [4]. DMOG under AO culture condition significantly increased ROS formation, likely through inhibition of PHD activity and induction of mitochondrial changes causing electron leaking from the respiratory chain [45]. It was shown that DMOG increased intracellular ROS production by decreasing mitochondrial membrane potential [42]. Hypoxia could upregulate a few genes that result in a redox potential shift by a switch from oxidative phosphorylation to glycolysis with an increase in NAD(P)H production [46].

We then measured mitochondrial burden in PC12 cells, which can be applied as a relative measure of mitochondrial mass. We found there was an increased mitochondrial burden (mass) in IH-cultured cells in comparison to AO-cultured cells, which may be related to changes in activating autophagy [47]. The DMOG-increased mitochondrial burden can be related to the production of inactive mitochondria [48]. Moreover, DMOG increased mitochondrial membrane potential after 72 h, which is consistent with several other studies [49,50,51]. The increase may have been related to inhibition of the mitochondrial F_0_ F_1_-ATPase leading to the slowing down of electron transport [52,53].

The cell cycle is an energy-demanding and very highly regulated process. The cell cycle consists of four distinct stages: gap phase 1 (G1), DNA replication phase (S), gap phase 2 (G2) and mitotic division (M). During cell division, cells switch between an oxidative phase which involves the biosynthesis of many cellular components (G1 phase), utilising energy derived from mitochondria, and a reductive phase with DNA replication and biosynthesis of mitochondria (S/G2/M phases) where energy is sourced from nonrespiratory energy production methods [54]. The studies of mechanisms that control mitochondrial biogenesis during the cell cycle have revealed that there is a synchronised increase in mitochondrial mass and membrane potential throughout the progression from G1 to mitosis and that these parameters return to normal after cell division [55]. If there is insufficient energy to complete the cycle, the cells become trapped in the G1 phase (restriction point) of the cell cycle. Analysis of the effects of hypoxia on the cell cycle showed that the CN culture condition reduced the S phase and elevated the G0-G1 phase with no significant effect on the G2-M phase. This is consistent with a few studies demonstrating that hypoxic cells were trapped in the G0/G1 phase of the cell cycle [47,56]. This may be related to the activation of protein phosphatase 2A (PP2A) and inhibition of extracellular signal-regulated kinase 1/2 (ERK1/2) phosphorylation which contribute to inhibiting PC12 proliferation through G0/G1 phase arrest [47]. On the other hand, DMOG trapped cells in the S phase. This may have been related to the induction of cell cycle arrest by the exertion of a markedly different pattern of regulation at various cell cycle checkpoint genes such as ataxia telangiectasia mutated, ataxia telangiectasia and Rad3 related (ATR), p53, p21, p27 and p21 [57]. The role of p27 is critical in cell cycle arrest under hypoxic conditions, although the exact role of HIF-1α in the induction of p27 is still an area of debate [58]. HIF-1α induction disrupts c-Myc–Max complex formation leading to decreased c-Myc transcription which induces p27 expression, causing cell cycle arrest [59,60]. In contrast, induction of HIF-2α can facilitate c-Myc and Max complex formation with increased c-Myc transcription which in turn increases *cyclin D2* expression and reduces p27, leading to cell proliferation [59,60]. Furthermore, HIFs play a role in the regulation of the transcription of several microRNAs, particularly miR210 which plays a role in regulating E2F3, a transcription factor causing severe downregulation in protein levels, leading to cell cycle progression arrest [61]. HIF-1α has the ability to block DNA replication under hypoxic conditions by activation of ATR and its downstream target checkpoint kinase 1, leading to cells being trapped in the G1 phase [62].

A core feature of PC12 cells is their neural differentiation capacity following exposure to the correct conditions. The differentiation could be achieved by encouraging adhesion of the PC12 cell suspension via the provision of a collagen IV coated substrate at low oxygen levels (1%, 4% and 12% O_2_) [63]. It was found that hypoxia (0.5% O_2_) not only caused rapid induction of neurite outgrowth but also synergistically enhanced NGF-induced neurite outgrowth up to 24 h [64]. We did not find significant neurite induction at 2% O_2_. DMOG was shown to induce PC12 cell differentiation and has no additive effect on PC12 differentiation over NGF in AO. This is consistent with a study that showed CoCl_2_ induced neurite outgrowth in PC12 cells [63].

In conclusion, hypoxia (2% O_2_) decreases the increase rate of PC12 cell numbers, reduces mitochondrial activity, stabilises HIF-1 and leads to cycle arrest at the G0/G1 phase compared to the AO culture condition. Both IH and DMOG induced immature morphology of mitochondria with lower metabolic activity, as determined by the lower mitochondrial mass and action potential. DMOG can mimic some of these effects of hypoxia, but it also induces other changes in the PC12 cells. DMOG is a nonspecific PHD inhibitor and could have dose-dependent side effects in some vital cellular processes in addition to HIF activation. The development of specific and highly potent small-molecule PHD inhibitors would be of great benefit for preserving the stemness of NSCs without inducing cellular toxicity. This is clinically relevant as a therapeutic approach for enhancing the number of NSCs in vitro for subsequent cell transplantation.

## Figures and Tables

**Figure 1 biomolecules-12-00541-f001:**
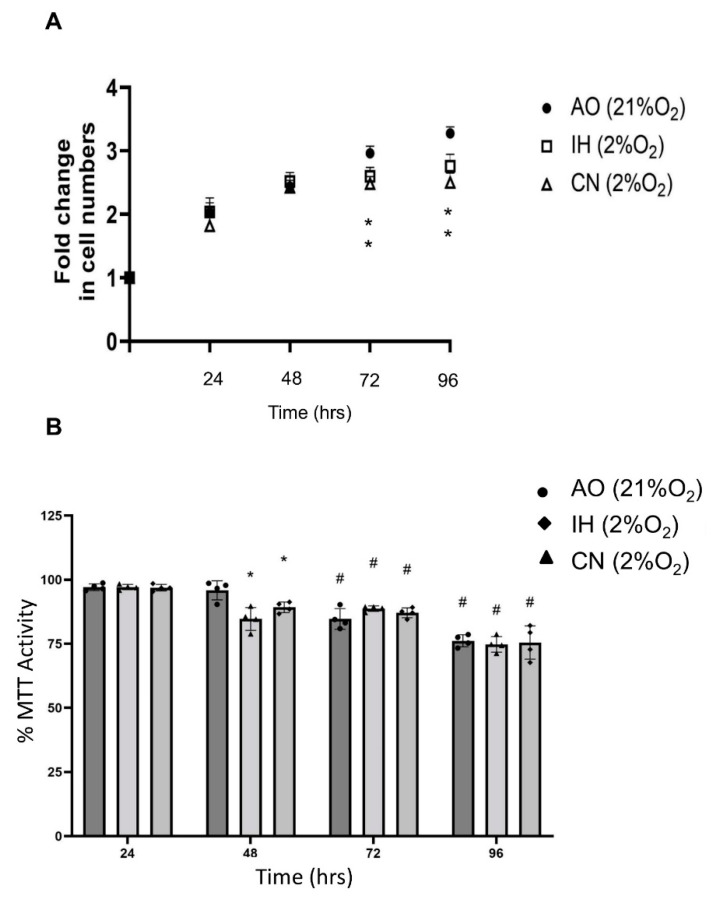
PC12 cell proliferation and MTT activities in different oxygen conditions. (**A**) Fold change in cell counts of PC12 cells under air oxygen (AO), intermittent hypoxia (IH) or continuous normoxia (CN) across a 96 h time-course. Cell proliferation was significantly reduced after 72 and 96 h incubation in both IH and CH. X-axis indicates time (h). Y-axis indicates fold changes in cell numbers. (**B**) MTT activity of PC12 cells in AO, IH or CH across a 96 h time-course. MTT activities were significantly reduced in both IH and CH compared to in AO at 48 h; however, there were no significant differences at other time points. In addition, the MTT activity values at 72 and 96 h were significantly lower than those at 24 h. X-axis indicates time (h). Y-axis indicates Abs. value normalised to Abs. of control at each time point. Data are expressed as mean + S.E.M., *n* = 4. Statistical significance of the assay was evaluated using one-way ANOVA with Tukey post hoc corrections; * indicates significant change in comparison to the AO control at each time point (*p* < 0.05). **^#^** indicates significant change in comparison to corresponding values recorded on day 1 (*p* < 0.05).

**Figure 2 biomolecules-12-00541-f002:**
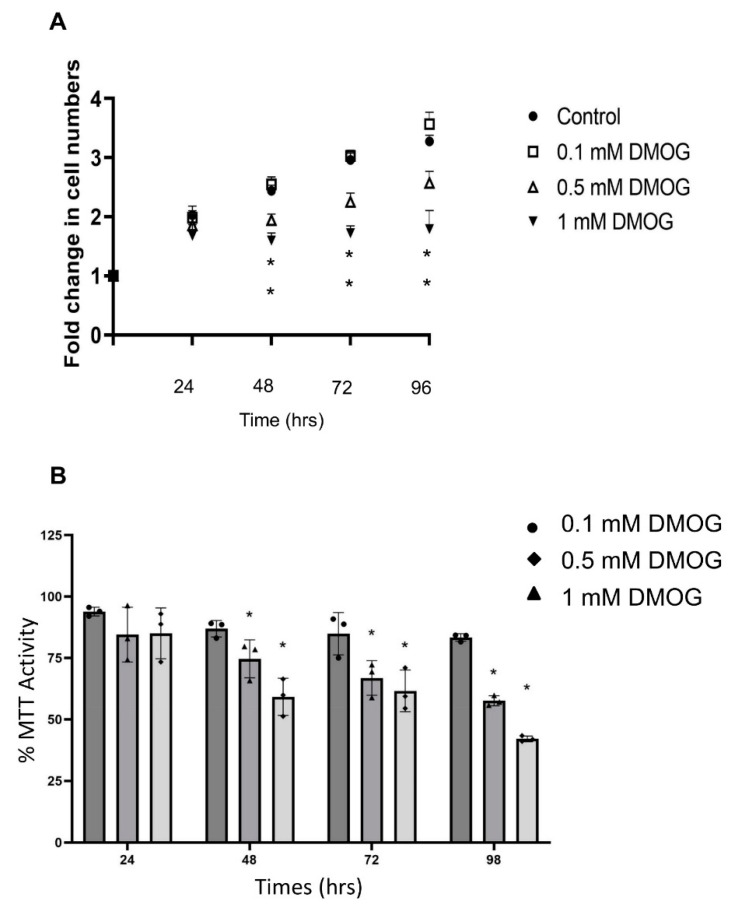
Effects of DMOG (0.1–1 mM) on PC12 cell proliferation and MTT activities. (**A**) Fold change in cell counts of PC12 cells under air oxygen (AO) following exposure to different concentrations of DMOG for a 96 h time-course. DMOG at 0.1 mM did not affect cell proliferation; however, DMOG at higher dose significantly inhibited cell proliferation at 48, 72 and 96 h. X-axis indicates time (h). Y-axis indicates fold changes in cell numbers. (**B**) MTT activity of PC12 cells in AO following exposure to different concentrations of DMOG across a 96 h time-course. MTT activities were significantly reduced by treatment with DMOG in a dose-related manner. X-axis indicates time (h). Y-axis indicates Abs. value normalised to Abs. of control at each time point. Data are expressed as mean + S.E.M, *n* = 3. Statistical significance of the assay was evaluated using one-way ANOVA with Tukey post hoc corrections. * indicates significant change in comparison to control at each time point (*p* < 0.05).

**Figure 3 biomolecules-12-00541-f003:**
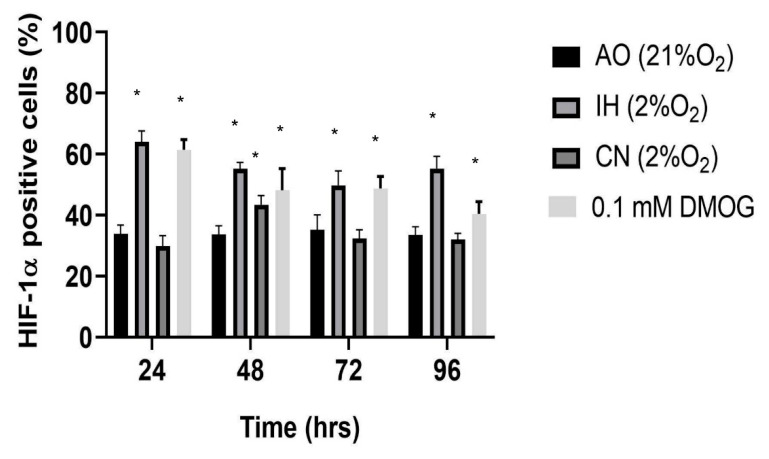
HIF-1α expression in different oxygen conditions and with DMOG. PC12 cells were incubated under air oxygen (AO), intermittent hypoxia (IH) or continuous normoxia (CN) or with DMOG (0.1 mM) over a 96 h time-course. The percentages of anti-HIF-1α (ab16066) labelled cells were significantly increased by the DMOG (0.1 mM) and IH compared to corresponding ones in AO; however, the CN did not change the HIF-1α expression significantly except at 48 h. Data are presented as % of HIF-1α positive cells ± S.E.M, *n* = 3. Statistical significance of the assay was evaluated using one-way ANOVA with Tukey post hoc corrections. * indicates significant difference in comparison to the positive cells at zero hour (*p* < 0.05).

**Figure 4 biomolecules-12-00541-f004:**
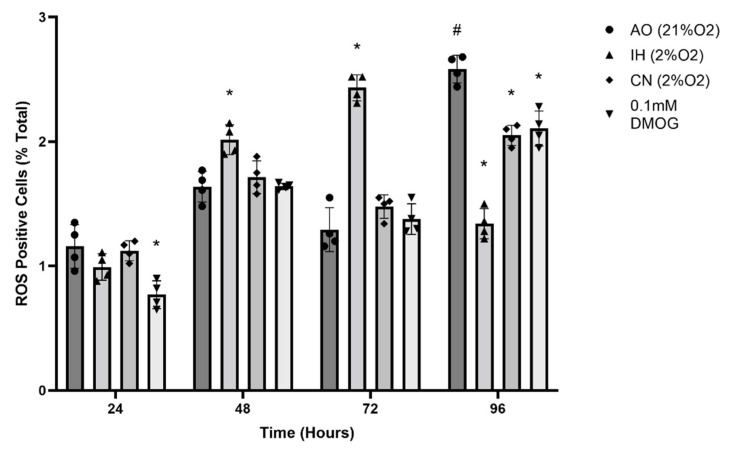
Reactive oxygen species (ROS) formation at different oxygen levels and with DMOG. Histogram of normalised fluorescence intensity of PC12 after stain with ROS-ID Hypoxia/Oxidative stress detection reagent. The exposure of PC12 cells to IH significantly increased the ROS level compared to cells in AO after 48 and 72 h. DMOG significantly reduced ROS formation at 24 h. Furthermore, cells in AO for 96 h significantly increased ROS formation compared to cells in AO after 24 h, while 2% O_2_ (IH and CN) and DMOG significantly reduced the increase in ROS formation levels after 96 h incubation. Data are presented as mean of normalised values ± S.E.M, *n* = 4. Statistical significance of the assay was evaluated using one-way ANOVA with Tukey post hoc corrections. * indicates significant difference in comparison to control at each time point (*p* < 0.05). ^#^ ndicates significant change in comparison to corresponding values recorded on day 1 (*p* < 0.05).

**Figure 5 biomolecules-12-00541-f005:**
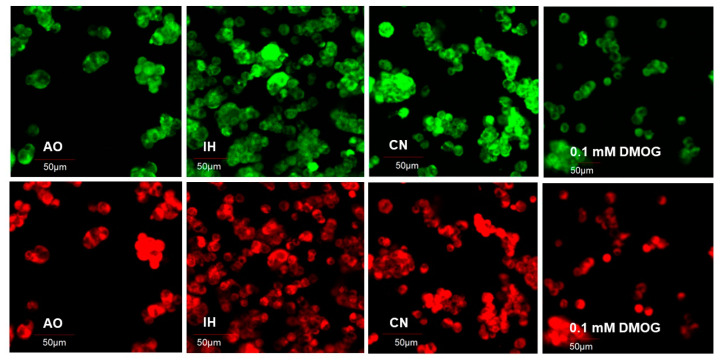
Representative images of PC12 cells loaded with MitoTracker Green and Red fluorescence at different oxygen levels and with DMOG. PC12 cells were cultured in collagen IV precoated 24-well plates in air oxygen (AO), intermittent hypoxia (IH) or continuous normoxia (CN) or with DMOG 0.1 mM in AO for a 96 h time-course. The top panels are MitoTracker Green fluorescence images; the bottom panels are MitoTracker Red fluorescence images. Scale bar indicates 50 µm.

**Figure 6 biomolecules-12-00541-f006:**
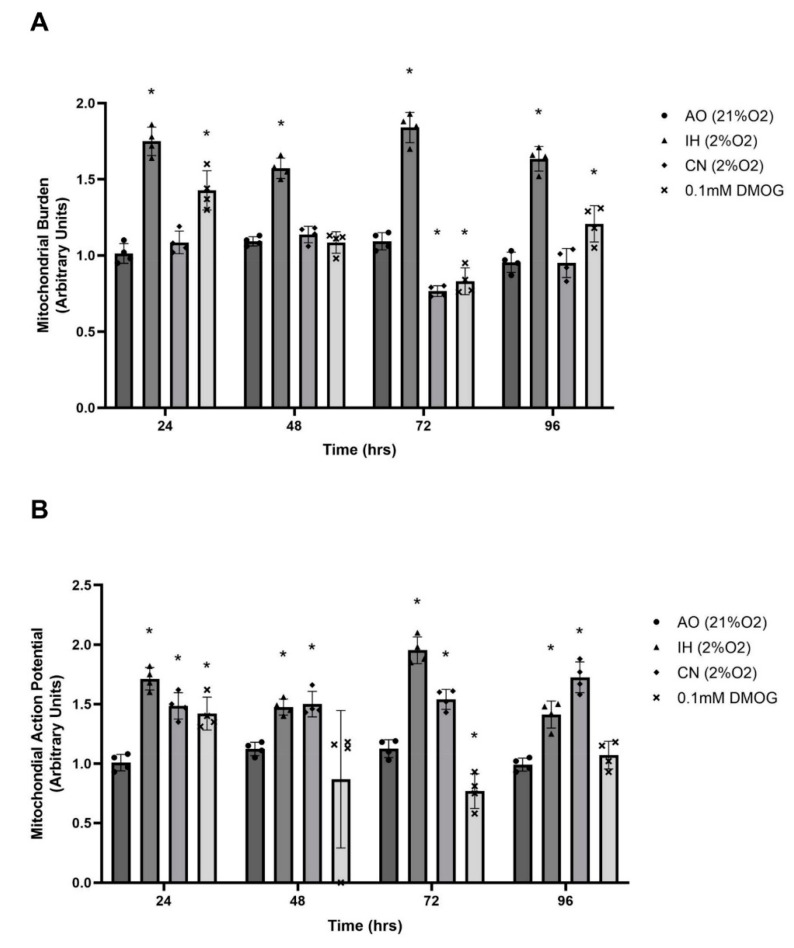
Mitochondrial burden and action potential of PC12 cells cultured at different oxygen levels and with DMOG. PC12 cells were cultured in air oxygen (AO), intermittent hypoxia (IH) or continuous normoxia (CN) or with DMOG (0.1 mM) in AO for a 96 h time-course. (**A**) MitoTracker Green fluorescence was measured by flow cytometry following labelling and incubation. Cells in IH had a significantly higher mitochondrial burden, while cells in CN and with DMOG had a similar mitochondrial burden compared to those in AO in the first 48 h, with a significant reduction at 72 h. (**B**) MitoTracker Red fluorescence was measured by flow cytometry following labelling and incubation. The mitochondrial action potentials were significantly increased under IH and CN incubation. DMOG significantly increased the mitochondrial action potentials at 24 h but reduced it at 72 h. Data are presented as normalised mean fluorescence intensity ± S.E.M, *n* = 4. Statistical significance of the assay was evaluated using one-way ANOVA with Tukey post hoc corrections. * indicates significant difference in comparison to AO at each time point (*p* < 0.05).

**Figure 7 biomolecules-12-00541-f007:**
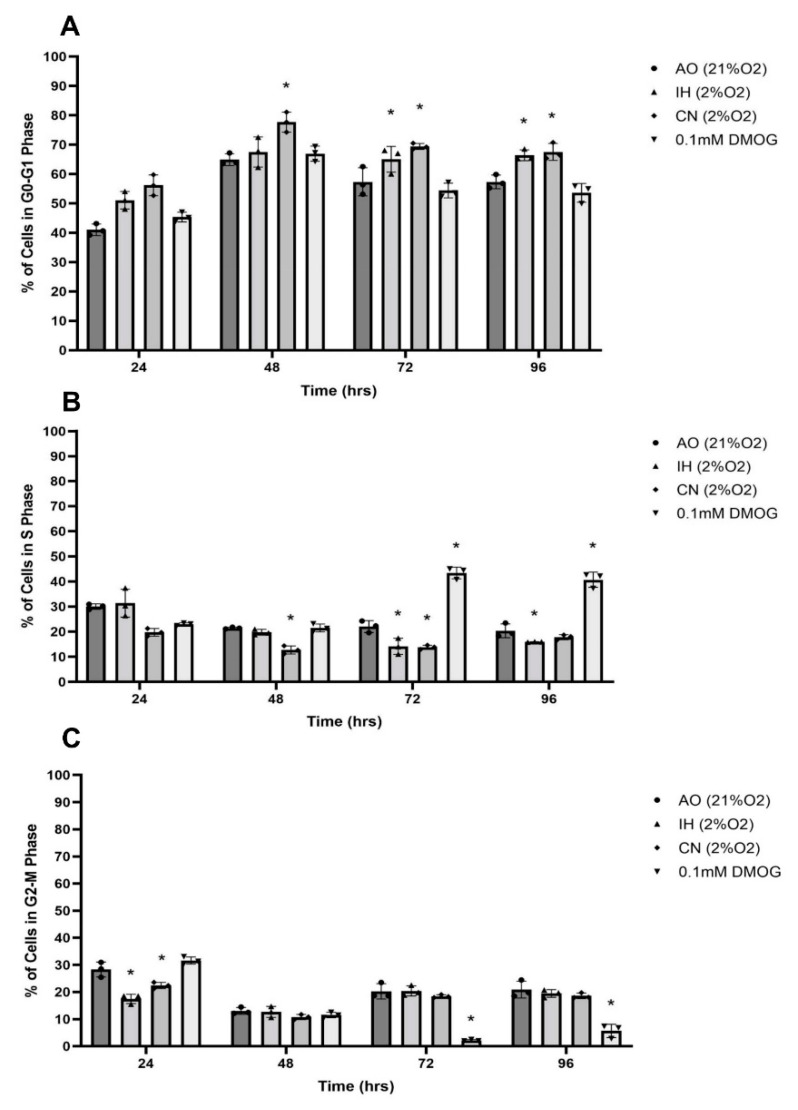
The cell cycle phases of PC12 cells cultured under three different culture conditions and with DMOG. PC12 cells were cultured in air oxygen (AO), intermittent hypoxia (IH) or continuous normoxia (CN) or with DMOG (0.1 mM) in AO for a 96 h time-course, and the percentage of cells in each cycle phase was analysed using a flow cytometer. (**A**) The percentage of cells in the G0/G1 phase was significantly increased in both IH and CN, while DMOG (0.1 mM) did not change it significantly. (**B**) The percentage of cells in the S phase was significantly reduced at 72 and 96 h in IH and CN, while DMOG (0.1 mM) significantly increased it at both 72 and 96 h. (**C**) The percentage of cells in the G2-M phase was significantly increased at 24 h in IH and CN; DMOG (0.1 mM) significantly reduced it at both 72 and 96 h. Data are presented as mean percentage of cells in a phase ± S.E.M, *n* = 3. Statistical significance of the assay was evaluated using one-way ANOVA with Tukey post hoc corrections. * indicates significant difference in comparison to control at each time point and under each oxygen culture condition (*p* < 0.05).

**Figure 8 biomolecules-12-00541-f008:**
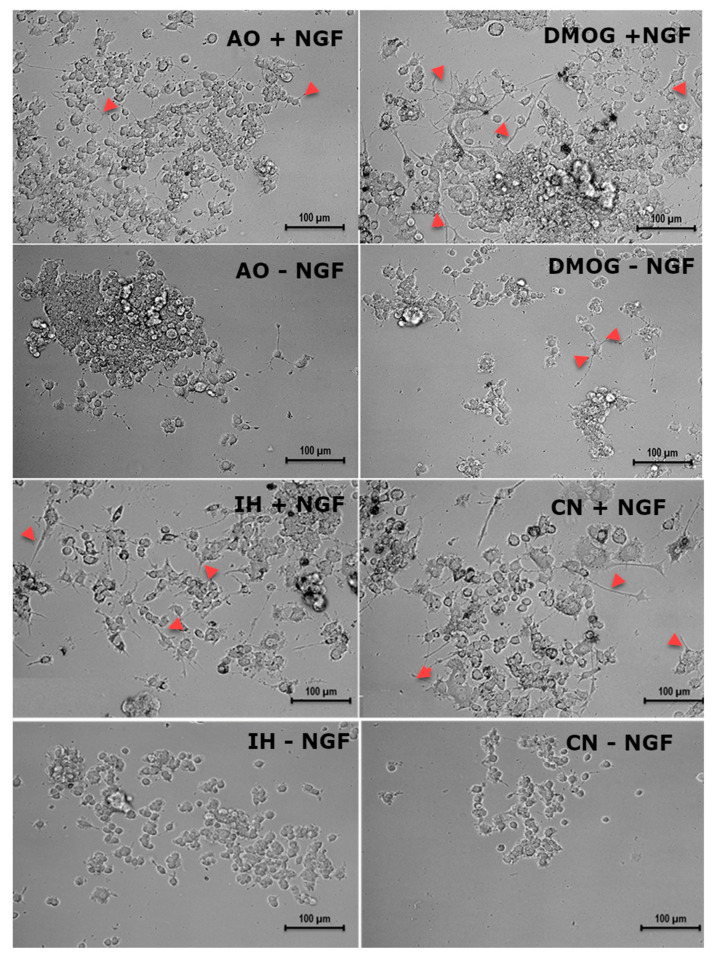
Representative images of PC12 cells at different oxygen levels and with DMOG, with or without nerve growth factor. PC12 cells were cultured in collagen IV precoated 24-well plates with and without nerve growth factor (NGF) stimulation for 7–10 days under three different oxygen culture conditions and with DMOG (0.1 mM). There was a significant increase in neurites with NGF treatment compared to cultures without NGF treatment, which had few neurites. On the other hand, DMOG also increased neurites under the air oxygen condition. The arrowheads indicate neurites. + NGF means with NGF; − NGF means without NGF. Scale bar indicates 100 µm.

**Figure 9 biomolecules-12-00541-f009:**
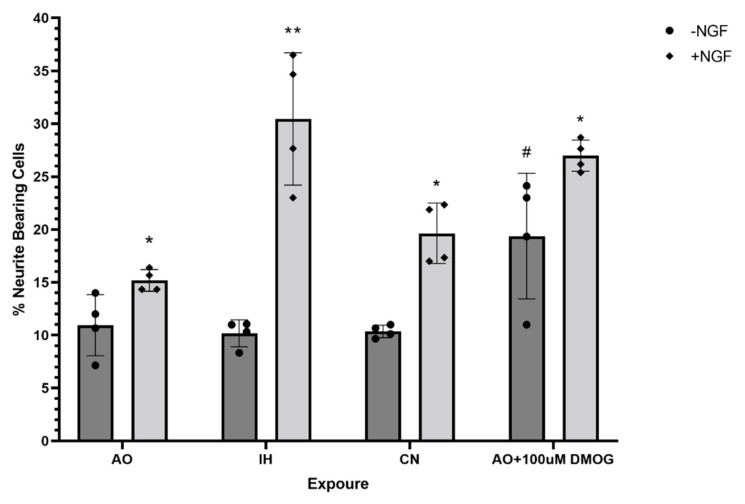
PC12 cell differentiation under three different culture conditions and with DMOG, with or without nerve growth factor. PC12 cells were cultured in collagen IV precoated 24-well plates with and without nerve growth factor (NGF) stimulation for 7–10 days under three different oxygen culture conditions and with DMOG (0.1 mM). The histogram shows the percentages of neurite-bearing PC12 cells. A significant increase in the percentage of neurite-bearing PC12 cells was shown with the treatment of NGF and DMOG (0.1 mM). Data are presented as mean percentage of cells in a phase ± S.E.M, *n* = 4. Statistical significance of the assay was evaluated using one-way ANOVA with Tukey post hoc corrections. * (*p* < 0.05) and ** (*p* < 0.01) indicate significant difference in comparison to cultures without NGF treatment. ^#^ indicates significant difference in comparison to culture in AO without NGF treatment (*p* < 0.05).

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
