# Peer review of "Dimethyloxalylglycine (DMOG), a Hypoxia Mimetic Agent, Does Not Replicate a Rat Pheochromocytoma (PC12) Cell Biological Response to Reduced Oxygen Culture"

_biomolecules, 2022, doi:10.3390/biom12040541_

Round 1

Reviewer 1 Report

The manuscript entitled “Dimethyloxalylglycine (DMOG), a hypoxia mimetic agent, does not replicate a RAT Pheochromocytoma (PC12) cell biological response to reduced oxygen culture” describes a study to investigate whether there is a difference between DMOG and reduced oxygen level in PC12 cells in a cell culture model. In this context, the authors showed that reduced oxygen levels did not affect PC12 cells in the same manner as the used hypoxia memetic agent DMOG.

Basically, the paper is presented in a logical manner. However, some issues appeared and the paper would benefit if these issues are solved.

Specific comments:

  • Fig.1: A decrease in the number of cells does not necessarily lead to a decrease in cell proliferation. In order to make such a statement, the authors should include BrdU incorporation. Do the cells go into apoptosis?
  • Fig.3: Under normal air conditions, HIFs are constantly degraded via the proteasome. It is therefore very surprising that the basal HIF-1α levels and especially the HIF-2α levels are so high. Can the authors explain that? Are there changes in protein synthesis under these conditions and time points?
  • What about HIF-1α and HIF-2α target genes?
  • Fig.4: The measurement of ROS species is very difficult due to their rapid reactivity. Therefore, the authors have to add both a positive and a negative control to the measurements. It is also very surprising that the basal values of ROS are so high under air conditions.
  • Fig.6: To make it visually more pleasant for the reader, I would recommend the authors to make 1 out of 3 graphs so that the reader can immediately see whether cells get somehow stuck in a phase or not.

Author Response

Specific comments:

Figure1: A decrease in the number of cells does not necessarily lead to a decrease in cell proliferation. In order to make such a statement, the authors should include BrdU incorporation. Do the cells go into apoptosis?

The reviewer is correct in their assertion that decreased numbers of cells does not automatically correlate with a decrease in cell proliferation. If apoptosis is happening at a greater rate than proliferation then it could mask continuing proliferation. We saw no evidence of apoptosis in data reported in 1A via high levels of trypan blue exclusion indicating consistent viability across all conditions. Figure 1 B measures MTT reduction which is a measure of mitochondrial metabolic activity and not proliferation. We are now using “reduction in the increase of cell numbers over the time” throughout the manuscript, rather than “a decrease in cell proliferation”.

Figure3: Under normal air conditions, HIFs are constantly degraded via the proteasome. It is therefore very surprising that the basal HIF-1α levels and especially the HIF-2α levels are so high. Can the authors explain that? Are there changes in protein synthesis under these conditions and time points?

HIFs are rapidly degradable proteins. The most common methods for detecting HIF alpha proteins include immunohistochemistry, flow cytometry, ELISA or Western blotting. Flow cytometry has been found to be more sensitive than visual immunofluorescence microscopy for the detection of ultralow levels of protein and may facilitate correct identification of negative cells when moderate or high levels of protein are present in a significant portion of cells (Ramberger et al., 2015; Tran et al., 2017).  

We used flow cytometry to detect the percentage of HIF positive cells. This does not mean high levels of HIF in the cells.

HIF stabilization by DMOG is through inhibition of prolyl hydroxylase, rather than changes in protein synthesis.

Ramberger M, Peschl P, Schanda K, Irschick R, Höftberger R, Deisenhammer F, Rostásy K, Berger T, Dalmau J, Reindl M. Comparison of Diagnostic Accuracy of Microscopy and Flow Cytometry in Evaluating N-Methyl-D-Aspartate Receptor Antibodies in Serum Using a Live Cell-Based Assay. PLoS One. 2015;10(3):e0122037. 

Tran DN, Smith SA, Brown DA, Parker AJ, Joseph JE, Armstrong N, Sewell WA. Polychromatic flow cytometry is more sensitive than microscopy in detecting small monoclonal plasma cell populations. Cytometry B Clin Cytom. 2017; 92(2):136-144.

What about HIF-1α and HIF-2α target genes?

We have studied HIF-1α and HIF-2α target genes expression by DMOG, and published the results in a separate study (Singh et al., 2020).

Figure4: The measurement of ROS species is very difficult due to their rapid reactivity. Therefore, the authors have to add both a positive and a negative control to the measurements. It is also very surprising that the basal values of ROS are so high under air conditions.

Elevated ROS levels in hyperoxic air culture are not surprising given the increased oxygen saturation levels (Stuart et al., 2018). The aim of this experiment was to evaluate the impact of alternate conditions on ROS production and where DMOG sat in this spectrum. The negative controls in this sense are already present via the untreated cells present. We used flow cytometry to detect the percentage of ROS positive cells.

Stuart JA, Fonseca J, Moradi F, Cunningham C, Seliman B, Worsfold CR, Dolan S, Abando J, Maddalena LA. How Supraphysiological Oxygen Levels in Standard Cell Culture Affect Oxygen-Consuming Reactions. Oxid Med Cell Longev. 2018 Sep 30;2018:8238459. 

Figure 6: To make it visually more pleasant for the reader, I would recommend the authors to make 1 out of 3 graphs so that the reader can immediately see whether cells get somehow stuck in a phase or not.

We understand the reviewers perspective but remain of the opinion that % cell number across the different phases gives the reader a whole picture of cell behavior in different conditions. Combining all data into a single graph would create a very complex visual presentation that would make interpretation more challenging.

Reviewer 2 Report

In the current study, authors were trying to study how PC12 cell line reacts to reduced oxygen culture. The major question is what is the significance of the current study and what is the rationale of the study? Please address the following questions.

  1. In Fig3, can authors explain by what method HIF-1a and HIF-2a expressions in each cell were measured? If measured by Flow Cytometry, can authors show data on how cells were gated and which cluster showing what percentage? Also, HIF-1a and HIF-2a mRNA expression and protein expression should be detected by qRT-PCR and immunoblotting in AO, IH, CN and DMOG treated cells.

  1. Again, in Fig4, if ROS was measured by Flow Cytometry, can authors show data on how cells were gated and which cluster showing what percentage? Please also use immunoblotting or staining to show the activation of ROS in differently treated cells.

  1. In Fig5A, B, some mitochondria markers should be tested either by immunoblotting or staining to show the changes in mitochondria functions in differentially treated cells.

  1. What is the rationale of this study? What is the significance of this study? What is the conclusion of this study?

Author Response

In the current study, authors were tr ying to study how PC12 cell line reacts to reduced oxygen culture. The major question is what is the significance of the current study and what is the rationale of the study? Please address the following questions.

For clarification the manuscript seeks to explore the fidelity of replication of reduced oxygen conditions by the hypoxia mimetic agent considered. The importance of this lies in not only elucidating if hypoxia mimetic agents truly replicate a hypoxia phenotype but also in potentially elucidating hypoxia independent actions of HMAs.

1. In Fig3, can authors explain by what method HIF-1a and HIF-2a expressions in each cell were measured? If measured by Flow Cytometry, can authors show data on how cells were gated and which cluster showing what percentage?

Also, HIF-1a and HIF-2a mRNA expression and protein expression should be detected by qRT-PCR and immunoblotting in AO, IH, CN and DMOG treated cells.

The reviewer is correct in their assumption that HIF expression was measured by FACS (as detailed in M&M Section 2.5). Forward and Side scatter were used to identify and gate cell populations before application of the relevant filter setting i.e., FL1.  FACS has greater sensitivity than immunoblot for the detection of protein arguing against the utility of including it for verification of FACS data. Further, irrespective of approach immunoblotting provides only semi-quantitative data vs. the fully quantitative data provided by FACS analysis. HIF stabilization by DMOG is through inhibition of prolyl hydroxylase, rather than changes in protein synthesis. HIF-1a and HIF-2a mRNA expression are not changed by hypoxia and DMOG (Singh et al., 2020).

2. Again, in Fig4, if ROS was measured by Flow Cytometry, can authors show data on how cells were gated and which cluster showing what percentage? Please also use immunoblotting or staining to show the activation of ROS in differently treated cells.

Measurement of ROS species is via chemical reaction with reactive species within cells and is not protein-based. Immunoblotting or staining are therefore not relevant in this setting. FACS settings are as described above.

3. In Fig5A, B, some mitochondria markers should be tested either by immunoblotting or staining to show the changes in mitochondria functions in differentially treated cells.

We have included representative images of both Mitotracker Green and Mitotracker Red to aid with interpretation (see our reply to reviewer 3 point 1). Neither Mitotracker dye lends itself to an immunoblot based investigation.

4. What is the rationale of this study? What is the significance of this study? What is the conclusion of this study?

We apologise if the rationale, significance, and conclusions are not readily apparent. The abstract details the rationale in the third sentence "The aim of this study is to determine whether chemical induced HIF accumulation mimics all aspects of the hypoxic response of cells." The rationale of this study is also in the last paragraph of the introduction (page4).

While the conclusion and significance are found in the final sentence of the abstract "However, DMOG does not provide an accurate replication of the reduced oxygen environments." These are also in the discussion first paragraph (page11) and last paragraph (page14).

Reviewer 3 Report

  1. Fluorescence images of cells loaded with probes (MitoTracker® Green and MitoTracker® Red) should be presented
  2. Predetermined sample size calculation. Specify if statistical methods were employed to predetermine the sample size and include a description of sample size calculations (provide all parameters of the calculation and estimation of effect size) in the manuscript. 
  3. Show individual data points (mandatory for small sample sizes n<15) as a dotplot or use box-plots instead of simple bar graphs. More information as to why this is important can be found here: https://journals.plos.org/plosbiology/article?id=10.1371/journal.pbio.1002128
  4. The work should discuss the mechanisms of ROS production in mitochondria and cytosol. Effects ischemia and preconditioning on gene expression. For example: https://pubmed.ncbi.nlm.nih.gov/34445509/, https://pubmed.ncbi.nlm.nih.gov/32219700/

Author Response

1. Fluorescence images of cells loaded with probes (MitoTracker® Green and MitoTracker® Red) should be presented

We have included a figure with fluorescence images of cells loaded with probes (MitoTracker® Green and MitoTracker® Red).

2. Predetermined sample size calculation. Specify if statistical methods were employed to predetermine the sample size and include a description of sample size calculations (provide all parameters of the calculation and estimation of effect size) in the manuscript. 

We did not employ sample size calculation in the conducting of our study. We utilised a well-established cell line as a model and included sufficient technical and biological replicates, based on our experience, to determine if changes reached statistical significance.

3. Show individual data points (mandatory for small sample sizes n<15) as a dotplot or use box-plots instead of simple bar graphs. More information as to why this is important can be found here: https://journals.plos.org/plosbiology/article?id=10.1371/journal.pbio.1002128

Thanks for the suggestion. We have changed.

4. The work should discuss the mechanisms of ROS production in mitochondria and cytosol. Effects ischemia and preconditioning on gene expression. For example: https://pubmed.ncbi.nlm.nih.gov/34445509/, https://pubmed.ncbi.nlm.nih.gov/32219700/

Agree, we have added them in the revised manuscript as follows:

“The mitochondria is the main source of endogenous ROS, but cytosolic enzymes are another source of ROS in cells (Balaban et al; 2005; Chen et al., 2018: Turovsky et al., 2021).”

“It has been now established that hypoxia increases ROS through transmission of electrons from ubisemiquinone to molecule oxygen at the Qo site of complex III of mitochondrial transport chain (Belle et al., 2007). Ischemia is also known to induce generation of ROS by both mitochondria (Abramov et al., 2007) and cytosolic enzymes (Turovasjya et al., 2019). Hippocampal neurons are characterized by high basic production of ROS, and ROS production in neurons increases rapidly in oxygen and glucose deprivation (Turovsky et al. 2021).”

Balaban RS, Nemoto S, Finkel T. Mitochondria, oxidants, and aging. Cell. 2005 Feb 25;120(4):483-95. doi: 10.1016/j.cell.2005.02.001. PMID: 15734681.

Bell EL, Klimova TA, Eisenbart J, et al. The Qo site of the mitochondrial complex III is required for the transduction of hypoxic signaling via reactive oxygen species production. J Cell Biol. 2007;177(6):1029-1036. doi:10.1083/jcb.200609074

Abramov A.Y., Scorziello A., Duchen M.R. Three distinct mechanisms generate oxygen free radicals in neurons and contribute to cell death during anoxia and reoxygenation. J. Neurosci. 2007;27:1129–1138.

Turovskaya M.V., Gaidin S.G., Mal’tseva V.N., Zinchenko V.P., Turovsky E.A. Taxifolin protects neurons against ischemic injury in vitro via the activation of antioxidant systems and signal transduction pathways of GABAergic neurons. Mol. Cell. Neurosci. 2019;96:10–24.

Turovsky EA, Varlamova EG, Plotnikov EY. Mechanisms Underlying the Protective Effect of the Peroxiredoxin-6 Are Mediated via the Protection of Astrocytes during Ischemia/Reoxygenation. Int J Mol Sci. 2021;22(16):8805. doi: 10.3390/ijms22168805.

Round 2

Reviewer 1 Report

The authors addressed the comments of the previous review and improved the quality of the manuscript.

Reviewer 2 Report

Authors have addressed all my concerns. 

Reviewer 3 Report

The article has undergone a serious and very professional revision. I am satisfied with the current version of the article and recommend it for publication.